# Preparation of Electrospun Active Molecular Membrane and Atmospheric Free Radicals Capture

**DOI:** 10.3390/molecules24173037

**Published:** 2019-08-21

**Authors:** Guoying Wang, Ying Su, Jianglei Yu, Ruihong Li, Shangrong Ma, Xiuli Niu, Gaofeng Shi

**Affiliations:** 1School of Petrochemical Engineering, Lanzhou University of Technology, Lanzhou 730050, China; 2Gansu Province Food Inspection Institute, Lanzhou 730050, China

**Keywords:** electrospun films, natural active molecules, free radicals capture, electrochemistry

## Abstract

We load the natural active molecules onto the spin film in an array using electrospinning techniques. The electrospun active molecular membranes we obtain in optimal parameters exhibit excellent capacity for scavenging radical. The reaction capacity of three different membranes for free radicals are shown as follow, glycyrrhizin acid membrane > quercetin membrane > α-mangostin membrane. The prepared active molecular electrospun membranes with a large specific surface area and high porosity could increase the interaction area between active molecules and free radicals. Additionally, it also has improved anti-airflow impact strength, anti-contaminant air molecular interference ability, and the ability to capture free radicals.

## 1. Introduction

Atmospheric free radicals are molecules, ions, or groups with unmatched electrons which have strong oxidizing capacity. They are the most important chemical intermediates or media in the atmosphere and play a crucial role in the atmospheric photo-chemical reactions. They determine the oxidation capacity of polluted air, in turn endangering human health and leading to a series of environmental problems [1,2,3]. Environmental toxicology research shows that free radicals in the urban atmosphere are a potential source which make humans more susceptible to developing cardiovascular, respiratory, and other diseases [4]. 

The natural active molecules found in the roots and leaves of plants and fruits, such as flavonoids, are a kind of polyphenol compounds in nature. Furthermore, they have important use value in pharmaceutical, industrial, and other industries [5,6]. The chemical structure of flavonoids consists of two benzene rings (ring A and ring B) which are connected by three carbon atoms. Most of them contain phenolic hydroxyl groups in their molecular structure; the phenolic hydroxyl group of flavonoids is the site of its antioxidant activity [7]. The phenolic hydroxyl group can provide hydrogen for the atmosphere free radical and then itself transform into a phenolic radical. This occurs under the resonance of the aromatic conjugated system forming semiquinone radical [8,9,10]. Therefore, they have a wide range of pharmacological activities, including anti-oxidation, anti-cancer, and anti-inflammatory properties, as well as helping with the recovery of injured tissues [11,12,13].For example, quercetin [14,15]; glycyrrhizic acid [16], mangosteen peel extract [17], etc., have been proven to have strong antioxidant capacity, free radical scavenging ability, and various pharmacological activities. 

In recent years, there have been several methods developed to be applied for atmospheric free radicals capture, such as salicylic acid impregnated membrane capturing atmospheric OH radicals, spin trapping technology, and flavonoid trap of electrospinning technology. In comparison, electrospinning technology has been viewed as the most versatile and effective technique for the preparation of continuous nanofibers with controlled morphology, structure and functional composition. The preparation of electrospinning active molecular membrane and applying it to capture atmospheric free radicals has rarely been reported. Gao et al. studied a low filtration resistance three-dimensional composite membrane via free surface electrospinning for effective PM_2.5_ capture [18]. Lin et al. studied electro spinning polyacrylonitrile–ionic nanofibers with superior PM_2.5_ capture capacity [19]. The electrospinning technique involves the application of a high-voltage electrostatic field to overcome the surface tension of droplets. The polymer charged solution is subjected to high-voltage static electricity. As the solvent evaporates while electrospinning, the jet will be elongated by electrostatic repulsion, and finally it is deposited on the collector to form a fiber mat similar to non-woven fabric [20,21]. Morphologies of the electrospun fibers could be affected by the following parameters: (a) the physical properties of spinning solution (e.g., concentration [22], viscosity, rheological [23] and conductivity [24]); (b) the operating parameters of the electrospinning instrument (e.g., voltage, flow rate [25], metal nozzle diameter and the distance between tip of the needle and collector [26]); (c) ambient conditions (e.g., temperature, relative humidity, etc.) [27,28,29].Through a series of screening experiments, we can obtain electrospun micro/nanofibers with ideal morphology, diameter, and minimal string-up bead formation. 

In this paper, three kinds of natural active molecules (quercetin, glycyrrhizic acid and mangosteen peel extract) were prepared into nanofiber membrane by using electrospinning technology. The active molecules were uniformly distributed and loaded into the nanofiber membrane in an array state by electrostatic action. The microstructure of nanofiber membrane was characterized by using scanning electron microscopy (SEM). The nanofiber membrane was used for atmospheric free radicals scavenging, and then evaluated by ultraviolet spectrophotometer, fluorescence spectrophotometer, and electrochemistry to select the molecular membrane with the best trapping ability. The nanofiber membrane with large specific surface area, high porosity and active molecule array distribution can be used for the capture, monitoring and scavenging of free radicals under heavy pollution conditions.

## 2. Results and Discussion

### 2.1. Impact of Factors on Active Molecular Membranes by Electrospinning

When working with electrospinning, two main parameter classes must be considered: the polymer addition amount and the processing conditions. In this research, considering to the numerous variables involved, we designed a series of single factor experiments to find the optimum working conditions. Table 1 reported the range of the experimental conditions [30].

#### 2.1.1. Polymer Addition Amount 

Polyvinylpyrrolidone (PVP) is an essential polymer for electrospinning. When the additional amount wasn’t enough, the filament cannot be formed. However, excessive PVP addition amount would cause the thickness of the spinning film to affect the air permeability.

According to the preliminary experiment, the PVP addition amount was 0.10 g, 0.15 g and 0.20 g in spinning solution, respectively [31]. As shown in Figure 1, when the amount of added PVP increased from 0.10 g to 0.20 g, the clearance rate also increased. As over 0.20 g, the spun film became thicker and it had so poor gas permeability that it had an adverse influence on the instrument. So the optimal PVP addition for quercetin was 0.20 g. When the amount of PVP was added to glycyrrhizic acid from 0.10 g to 0.20 g, the clearance rate was at its highest at 0.15 g. Therefore, the optimal PVP addition for glycyrrhizic acid was 0.15 g. As the PVP addition increased from 0.10 g to 0.20 g, the clearance rate decreased at the beginning and then increased. When the addition amount was less than 0.10 g, the filament cannot be formed. Therefore, the optimal PVP addition for α-mangostin was 0.10 g. Table 2 shows the average diameter of electrospun fibers at various PVP addition amounts. Diameters of the spun fibers were measured directly by corresponding measurement software in various SEM images taken for each specimen, from which at least 20 measurements for each specimen were analyzed to obtain an average value as well as the standard deviation. As shown in Table 2, the diameter of fiber increased as the PVP concentration increased. When the PVP addition amount was 0.10 g, 0.15 g, and 0.20 g, the diameter of the fiber we obtained was 0.11 μm, 0.25 μm and 0.43 μm, respectively.

#### 2.1.2. Voltage

The higher the electro-spinning voltage was, the thinner the spun filament was; this would affect the film formation and property. But as the voltage was too low, the liquid could overcome its own gravity and therefore cause droplet formation.

According to the pre-experiment, the voltages of 10 kV, 12 kV and 16 kV [32,33] were selected for screening. As can be seen from Figure 2, when the voltage for quercetin increased from 10 kV to 16 kV, the clearance rate also increased. But above 16 kV, the membrane was relatively thin and excessive voltage could easily create a dangerous situation. Therefore, the optimal voltage for quercetin voltage was 16 kV. When the voltage for glycyrrhizic acid increased from 10 kV to 16 kV, the clearance rate was the highest at 12 kV. So the optimal voltage of glycyrrhizic acid was 12 kV. With the increase of the voltage for α-mangostin from 10 kV to 16 kV, the clearance rate increased; but over 16 kV, the spun film was too thin, and the voltage was too high to generate danger. Therefore, the optimal voltage for α-mangostin was also 16 kV. Table 3 showed the average diameter of electrospun fiber at various levels of voltage, it was shown that as the applied voltage increased, the fiber diameter decreased. In this experiment, the average diameter of fiber increased from 0.51 μm to 0.32 μm as the applied voltage increased from 10 kV to 16 kV.

#### 2.1.3. Spinning Distance

If the spinning distance (distance between tip of the needle and collector) was too large, the electro-spinning liquid cannot overcome its own gravity, and droplets were easy to produce. If the receiving distance was too small, the spinning effect would be unsatisfactory.

According to the preliminary experiment, the distance between tip of the needle and collector were selected to be 10 cm, 15 cm, and 20 cm [34,35] for screening. It can be seen from Figure 3 that when the spinning distance of quercetin increased from 10 cm to 20 cm, the clearance rate also increased. But if it exceeded 20 cm, the liquid cannot overcome its own gravity, and it was easy to cause droplet formation. Therefore, the optimal distance of quercetin was 20 cm. When the spinning distance of glycyrrhizic acid increased from 10 cm to 20 cm, the removal rate was best at 15 cm. So the optimal distance of glycyrrhizic acid was 15 cm. As the spinning distance of α-mangostin increased from 10 cm to 20 cm, the clearance rate increased; however, if the spinning distance exceeded 20 cm, the electro-spinning liquid cannot become nanofiber and it became easy to create small droplets that fall down. Hence the optimum distance of α-mangostin was 20 cm. Table 4 showed the average diameter of the fibers prepared at different electrospinning distances. The average fiber diameter would decrease as the spinning distance increased.

#### 2.1.4. Injection Speed

Under the condition of a high pushing speed, the liquid cannot become nanofiber even in a high-potential electric-field. But if the pushing speed was too low, the spinning formation and its performance were not good.

According to the preliminary experiment, three different injection speed of 0.10, 0.18, and 0.26 mm/min [36,37] were used in this study. It can be seen from Figure 4 when the pushing speed for quercetin increased from 0.10 to 0.26 mm/min, the clearance rate also increased. But over 0.26 mm/min, the liquid cannot overcome its own gravity to become nano-fiber. Therefore, the optimal injection speed of quercetin was 0.26 mm/min. As the injection speed of glycyrrhizic acid increased from 0.10 to 0.26 mm/min, the clearance rate was the highest at 0.18 mm/min. So the pushing speed of glycyrrhizic acid was selected at 0.18 mm/min. When the injection speed of α-mangostin increased from 0.10 to 0.26 mm/min, the clearance rate also increased; but over 0.26 mm/min, the excessive liquid could not overcome its own gravity and become small droplets falling down. Above all, the optimal pushing speed of α-mangostin was selected 0.26 mm/min. Table 5 showed the influence on the average diameters of electrospun fibers after changing the flow rate of spinning solution to 0.10~0.26 mm/min. With the increase of the flow rate of spinning solution, the average diameter also increased.

According to the results of screening tests, we concluded the optimal spinning condition of quercetin; glycyrrhizic acid; α-mangostin shown in Table 6. The removal rate of the spun membrane in optimal spinning condition was the highest. 

### 2.2. Structural Characterization

#### Scanning Electron Microscopy (SEM)

The films were characterized by scanning electron microscopy (SEM). Figure 5 (A1,B1,C1) showed the morphological characteristics of quercetin, glycyrrhizic acid and α-mangostin spinning membrane at the optimal spinning conditions. Figure 5 (A2,B2,C2) showed the morphological characteristics of quercetin, glycyrrhizic acid, and α-mangostin spinning membrane after sampling.

It can be seen from Figure 5 (A1,B1,C1) that the three active materials on the electrospinning membranes did not agglomerate and uniformly distributed into a network structure, which increased the contact area between the radicals and the active material. Therefore, the unique structures made the removal effect better. Compared with ordinary films, the electrospinning film can improve the anti-interference ability of airflow and the free radical trapping effect was improved. Figure 5 A2,B2,C2 exhibited separately the morphology characteristics for three kinds of active material after sampling for 3 h. From the picture, we can clearly observe that active substances were uniformly distributed in an array before sampling while the structure of fibrous wire was destroyed and the agglomerate of active molecule was decreased after sampling. This indicated that the reaction between active substance and hydroxyl free radical happened.

### 2.3. Detection Method

#### 2.3.1. UV Absorption Spectrum Detection

The absorbance of the three molecules at the maximum absorption peak was determined, and due to the different absorbance before and after the reaction of molecules we concluded the clearance rate, which was calculated by formula (1.1), and the free radical concentration was calculated by formula (1.2). The clearance rate of the pristine membrane (only PVP without any active molecule) was 0%.

Figure 6 showed the clearance rate of the three active molecules before and after sampling of at the same concentration, under the same spinning condition and in the same sampling environment. It can be seen from the figure that all the three active molecules have remarkable scavenging effects on free radicals, and the scavenging abilities of the three active substances on free radicals in the atmosphere is ranked: glycyrrhizic acid > quercetin > α-mangostin.

As shown in Figure 7, the UV curves for nanofiber membrane with three active molecules after and before sampling display consistent variation characteristics. The UV absorbance of three kinds of active substances after sampling started to decrease; there was no peak type change and new peaks would appear. In the process of the active substance and free radicals reaction, free radicals would react with phenolic hydroxyl groups on the benzene ring, strip the hydrogen ions on the phenol hydroxyl and generate H_2_O, ROOH etc. After being striped the hydrogen, phenolic hydroxyl was affected by the resonance of aromatic ring conjugate system and generated stable half quinone free radicals. This reaction terminated free radical chain reaction, at the same time n-π* transition electronic decreased significantly. The loss of active molecules leaded to the decrease of absorbance, so the active substance has a scavenging effect on free radicals. The precision of the method was evaluated by performing six to nine assays of the test samples and calculating the RSD %. The results are shown in Table 7. Table 8 showed the environmental parameters and free radical concentration around the sampling site.

#### 2.3.2. Electrochemical Detection

Figure 8 showed the cyclic voltammetry curves of the reaction between the three active substances and free radicals. It can be seen from the figure that the oxidation potential of the three active molecules remains within a certain range. When the free radicals react with the active molecules, redox reaction occurs, electron loss occurs, and the redox is weakened. The peak position of quercetin did not change before and after sampling, but the electron loss did occur after sampling. After the sampling with glycyrrhizic acid, the peak pattern was shifted and the electron loss was obvious. After sampling with α-mangostin, the peak type of α-mangostin did not change, but the peak area decreased, indicating that the active substance was consumed during the reaction with free radicals. The electrochemical reactions occur on the 4-carbonyl groups, and the process of generating free radicals is an irreversible electrode process which controls both the electrode reaction and the chemical reaction. The RSD % of the method is shown in Table 7.

#### 2.3.3. Fluorescence Emission Spectrometry

As shown in Figure 9, the three active molecule themselves had much weaker fluorescence signals. After reacting with hydroxyl radicals in the atmosphere, the additive product with fluorescence were produced. So the fluorescence intensity signal of the sample would significantly increase. Then we could conclude the clearance rate according to the variation of peak area generated before and after sampling. The maximum excited wavelengths of the samples were 360 nm, 350 nm and 391 nm, respectively. Table 7 shows the RSD % of the method.

## 3. Experimental Section

### 3.1. Three Natural Active Molecules 

Quercetin is the widely distributed flavonoid in the plant kingdom. Glycyrrhizic is a high-sweet sweetener and detoxification food and its extract (glycyrrhizic acid) has the ability of scavenging hydroxyl radical. A variety of molecules extractable from mangosteen peel, including mangostin and procyanidins etc., have been proved to have strong antioxidant capacity, free radical scavenging ability and various pharmacological activities. Their chemical structural formulas are shown in Table 9 and Figure 10.

### 3.2. Chemicals and Equipment

Polyvinylpyrrolidone (PVP) was purchased from Shanghai McLean Biochemical Technology Corp. (Shanghai, China). Absolute methanol (analytic reagent) was obtained from Shanghai Xingke high Purity Solvent Corp. (Shanghai, China).

Intelligent medium flow air total suspended particulate sampler was purchased from Wuhan TianHong Instrument Corp. (TH-150A, Wuhan, China). Electrostatic spinning equipment was purchased from Beijing Yongkang Leye Technology Development Corp. (NL18-20, Beijing, China). UV-VIS spectrophotometer was purchased from Thermo Fisher scientific. (GEN10S UV-Vis, Waltham, MA, USA). A fluorospectro photometer was purchased from PerkinElmer Corp. (LS55, Boston, MA, USA). A multichannel Electrochemical Tester was purchased from Princeton Technology. (Parsat MC, Berwyn, PA, USA).

### 3.3. Extraction of Active Material

The fresh oak bark, glycyrrhiza, and mangosteen peel were dried and pulverized, and a certain amount of three kinds of powder after dry were separately put into flask with round bottom with hot reflux extraction. The separation conditions are shown in Table 10. 

### 3.4. Electrospinning

Electrospinning was carried out at room temperature, First of all, PVP and active moleculars including quercetin of 0.04% (*w/v*), glycyrrhizic acid of 0.06% (*w/v*) and α-mangostin of 0.03% (*w/v*) was dissolved into 5 mL methanol, respectively; and then the spinning solution was stirred with a magnetic stirrer for 1 h and standing for 5 min. The prepared three kinds of spinning solution were placed in a 5 mL syringe. The metallic needle was connected to the positively-charged, and connected to the aluminum foil paper with glued three glass fiber membranes to grounded collector. In addition to adjusting the operating parameters of the electrospinning apparatus, the receiving speed was set at 40 r/min. Spinning parameters of the three active substances are shown in Table 11. Flow diagram of the preparation of electrospinning active molecular membrane are shown in Figure 11.

### 3.5. Free Radical Trapping

The sampling site was selected as the rooftop on the 15th floor of tall building of Lanzhou University of Technology. The prepared nanofiber membrane was placed in the intelligent medium-flow air total particulate sampler (TH-150A, Wuhan, China) for sampling for 3 h, and the flow rate was 100 L/min, while recording environmental parameters. And then the after sampling nanofiber membrane was dissolved in 10ml methanol, and transferred to a sample bottle and analyzed by the detection systems.

### 3.6. Detection Method

Ultraviolet absorption spectrum detection parameters: the PVP blank membrane dissolved in methanol was used as the background, the scanning range was from 190 to 1100 nm, and the experiment repeated 3 times. The characteristic absorption peak of quercetin was 370 nm [46], the characteristic absorption peak of glycyrrhizic acid was 252 nm [47], and the characteristic absorption peak of α-mangostin was 317 nm [48].

Fluorescence emission spectra detection parameters: with the change of excitation wavelength, we obtained the emission spectra of fluorescence substances varying from 200 to 800 nm. The wavelength of quercetin excitation was 360 nm, the wavelength of glycyrrhizin excitation was 350 nm, and the wavelength of alpha-mangostin excitation was 391 nm.

Electrochemical experimental parameters: The experiment used a three-electrode system with a calomel electrode as a reference electrode, a platinum wire electrode as an auxiliary electrode, and a bare glassy carbon electrode as a working electrode. In the condition of Na_2_HPO_4_-NaH_2_PO_4_ buffer system, 10 mL of Na_2_HPO_4_-NaH_2_PO_4_ buffer solution was decanted into the electrolytic cell. The quercetin solution, glycyrrhizic acid solution and α-mangostin solution were separately added to the buffer by a pipette and were allowed to stand. The material to be detected was enriched to the surface of the working electrode and was detected by cyclic voltammetry (CV). The scanning range was −2~2 V, the scanning cycle was 3, the quercetin sweep rate was 50 mv/s, and the glycyrrhizic acid sweep rate was 70 mv/s, α-mangostin sweep rate was 100 mv/s.

The clearance rate (reactivity) and free radical concentration calculation formula was:Clearance rate (%) = (A_0_ − A)/A_0_(1)
A_0_—Absorbance at the characteristic absorption peak before sample samplingA—Absorbance at the characteristic absorption peak after sample sampling
(2)Free radical concentration=Fx×K×N×NAFg×t×(1−α)
F_x_—Sample clearance (%)K—correction factor (1 × 10^−8^)N—The amount of substance in the sample (mol)N_A_—Avogadro’s number (6.02 × 10^23^)F_g_—The sampler sets the intake flow (cm^3^/min)t—Sampling time (min)α—The product loss rate (5%)

## 4. Conclusion

In this paper, active glycyrrhizic acid electrospinning membranes, active quercetin electrospinning membranes and active quercetin electrospinning membranes were prepared in our lab. The reaction capacity of three different membranes for free radicals are shown as follows: glycyrrhizic acid membrane > quercetin membrane > α-mangostin membrane. The free radicals detection were carried out by fluorescence emission spectroscopy, UV absorption spectroscopy, and the electrochemistry method, respectively. The prepared active molecular electrospinning membranes have large specific surface area and high porosity, which could increase the interaction area between active molecules and free radicals. The active molecular electrospinning membranes also have improved anti-airflow impact strength, anti-contaminant air molecular interference ability, and good ability to capture free radicals. Additionally, the active molecular electrospinning membranes are also suitable for carrying sampling by drones, which provides a new technical method for the study of free radicals and atmospheric oxidative spatial variation.

## Figures and Tables

**Figure 1 molecules-24-03037-f001:**
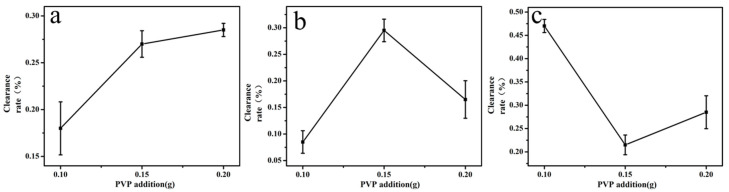
Impact of PVP addition amount on the clearance rate of active molecular membranes (**a**: quercetin; **b**: glycyrrhizic; **c**: α-mangostin).

**Figure 2 molecules-24-03037-f002:**
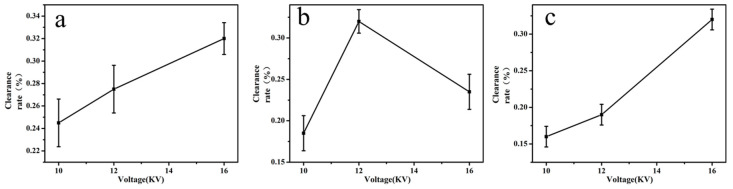
Impact of voltage on the clearance rate of active molecular membranes (**a**: quercetin; **b**: glycyrrhizic; **c**: α-mangostin).

**Figure 3 molecules-24-03037-f003:**
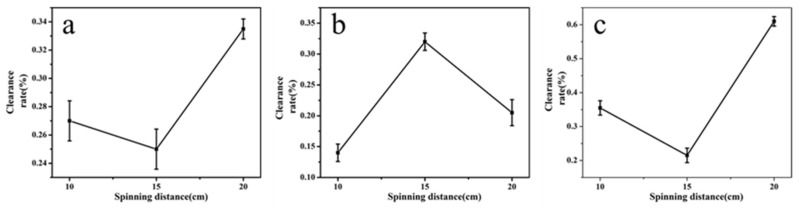
Impact of spinning distance on the clearance rate of active molecular membranes (**a**: quercetin; **b**: glycyrrhizic; **c**: α-mangostin).

**Figure 4 molecules-24-03037-f004:**
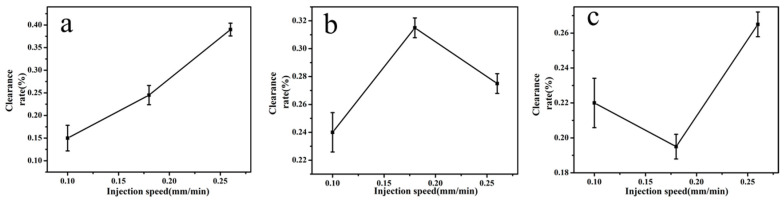
Impact of injection speed on the clearance rate of active molecular membranes (**a**: quercetin; **b**: glycyrrhizic; **c**: α-mangostin).

**Figure 5 molecules-24-03037-f005:**
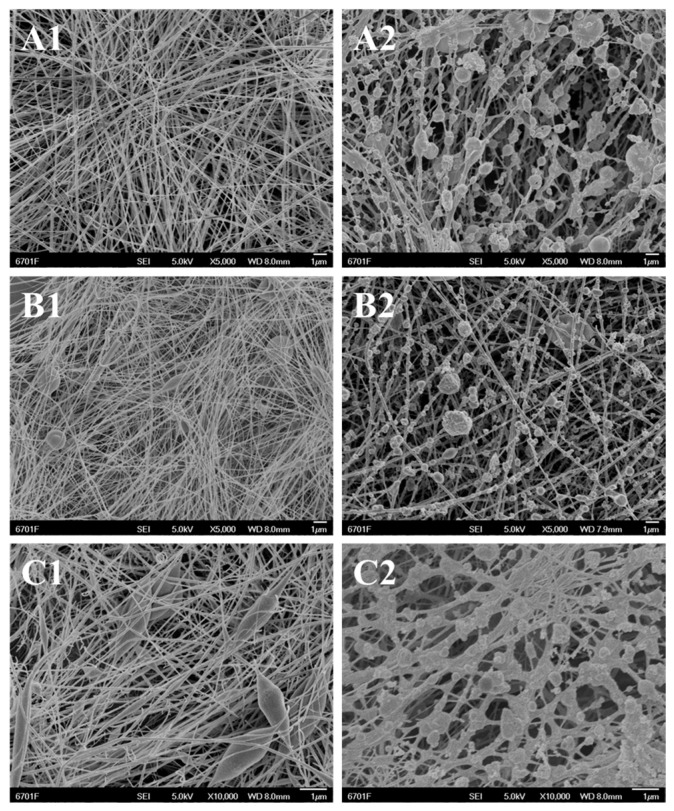
The SEM images of spinning membrane (**A1**: quercetin; **B1**: glycyrrhizic; **C1**: α-mangostin); and after sampling (**A2**: quercetin; **B2**: glycyrrhizic; **C2**: α-mangostin).

**Figure 6 molecules-24-03037-f006:**
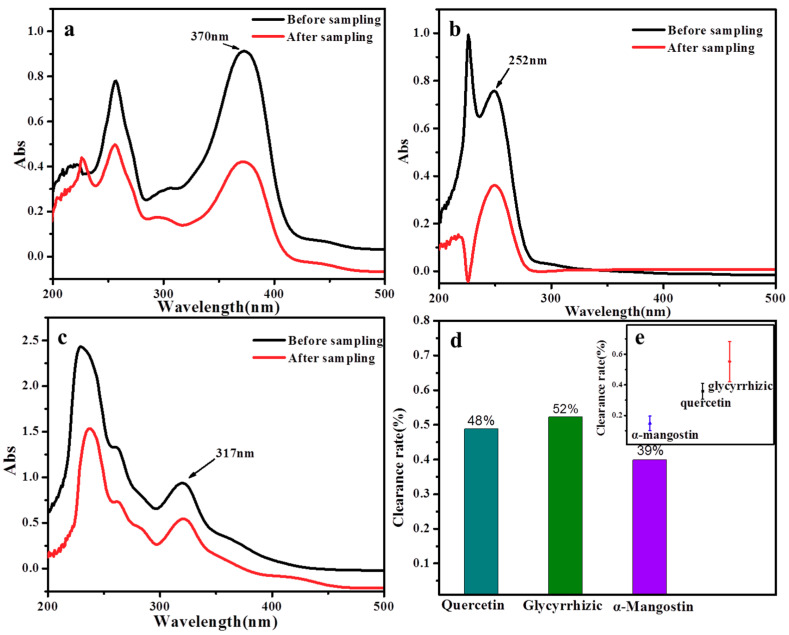
(**a**) UV curves for quercetin before and after sampling; (**b**) UV curves for glycyrrhizic before and after sampling; (**c**) UV curves for α-mangostin before and after sampling; (**d**) the clearance rate of three active molecules(%); (**e**)Error bars for the clearance rate of three active molecules.

**Figure 7 molecules-24-03037-f007:**
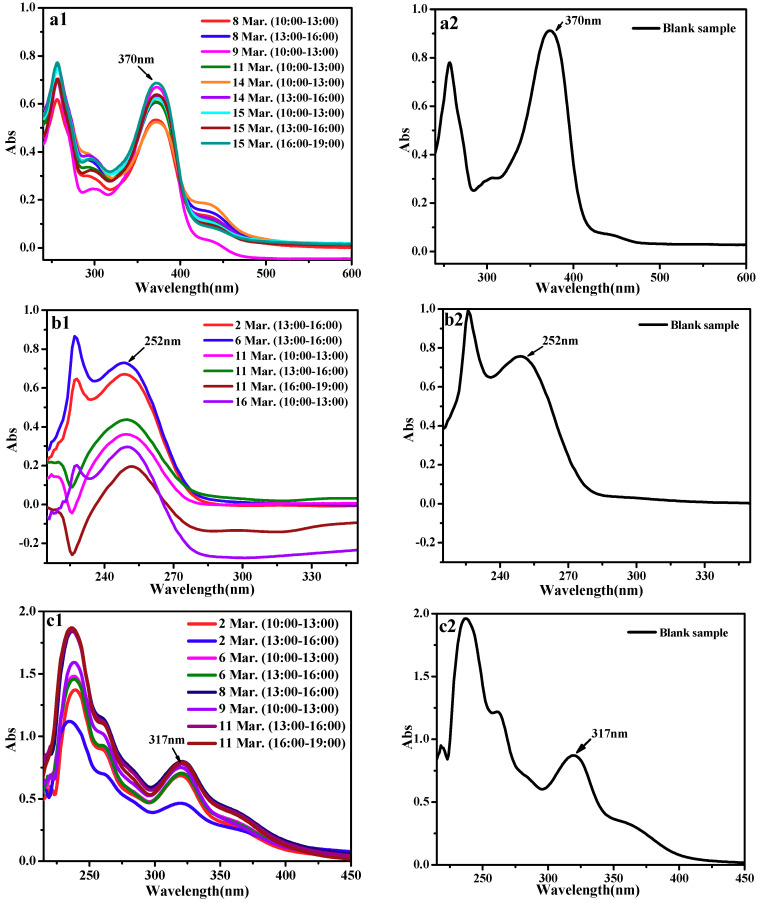
(**a**) UV curves for quercetin after and before sampling; (**b**) UV curves for glycyrrhizic after and before sampling; (**c**) UV curves for α-mangostin after and before sampling.

**Figure 8 molecules-24-03037-f008:**
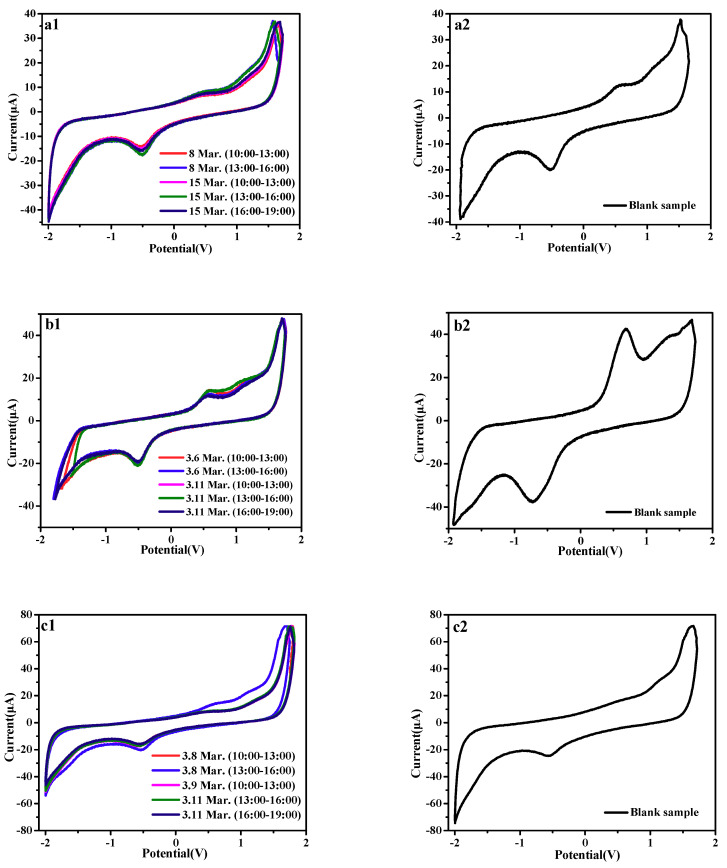
(**a**) CV curves for quercetin after and before sampling at a scan rate of 50 mv/s; (**b**) CV curves for glycyrrhizic after and before sampling at a scan rate of 70 mv/s; (**c**) CV curves for α-mangostin after and before sampling at a scan rate of 100 mv/s.

**Figure 9 molecules-24-03037-f009:**
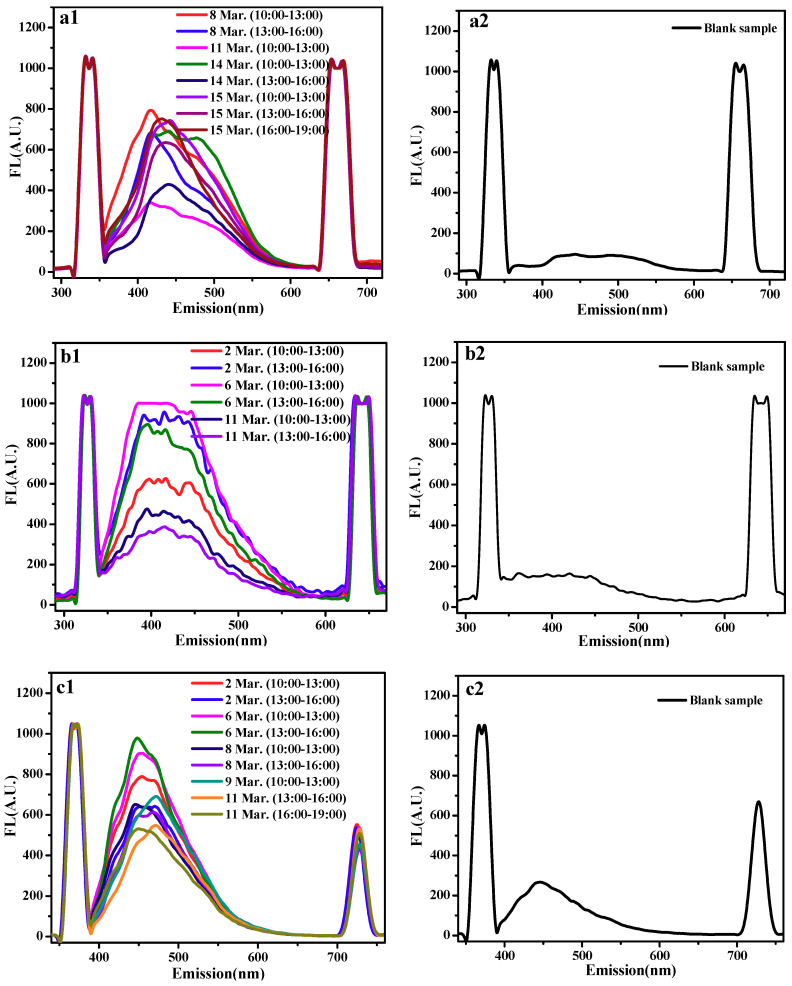
The fluorescence intensity variation of three molecules after and before sampling (**a**: quercetin, **b**: glycyrrhizic, **c**: α-mangostin).

**Figure 10 molecules-24-03037-f010:**
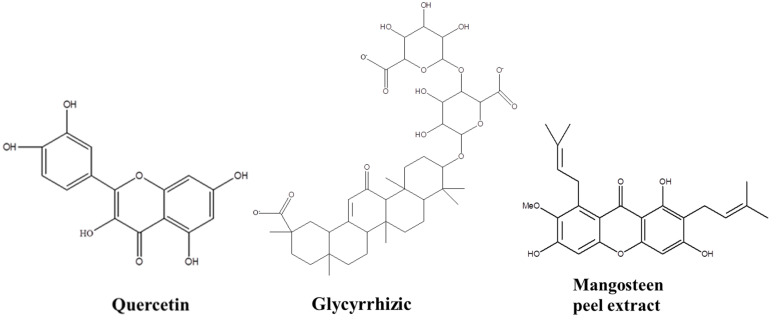
Chemical structure of three active molecules.

**Figure 11 molecules-24-03037-f011:**
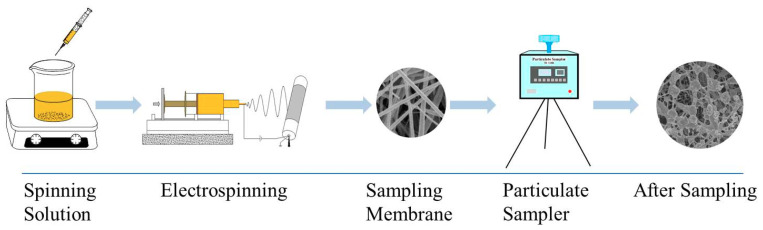
Flow diagram of the preparation of electrospinning active molecular membrane and application for capturing free radicals.

**Table 1 molecules-24-03037-t001:** The main range of the experimental conditions.

PVP Addition Amount (g)	Voltage (kv)	Spinning Distance (cm)	Injection Speed (mm/min)
0.10	10	10	0.10
0.15	12	15	0.18
0.20	16	20	0.26

**Table 2 molecules-24-03037-t002:** The range of fiber diameter with the different PVP addition amount.

PVP Addition Amount (g)	Voltage (kv)	Spinning Distance (cm)	Injection Speed (mm/min)	Fiber Aiameter (μm)
0.10	12	15	0.18	0.11 ± 0.04
0.15	12	15	0.18	0.25 ± 0.05
0.20	12	15	0.18	0.43 ± 0.17

**Table 3 molecules-24-03037-t003:** The impact of the voltage on the diameter of fiber.

PVP Addition Amount (g)	Voltage (kv)	Spinning Distance (cm)	Injection Speed (mm/min)	Fiber Diameter (μm)
0.15	10	15	0.18	0.51 ± 0.07
0.15	12	15	0.18	0.42 ± 0.09
0.15	16	15	0.18	0.32 ± 0.09

**Table 4 molecules-24-03037-t004:** The impact of the spinning distance on the diameter of fiber.

PVP Addition Amount (g)	Voltage (kv)	Spinning Distance (cm)	Injection Speed (mm/min)	Fiber Diameter (μm)
0.15	12	10	0.18	0.61 ± 0.15
0.15	12	15	0.18	0.41 ± 0.11
0.15	12	20	0.18	0.35 ± 0.17

**Table 5 molecules-24-03037-t005:** The impact of the injection speed on the fiber diameter.

PVP Addition Amount (g)	Voltage (kv)	Spinning Distance (cm)	Injection Speed (mm/min)	Fiber Diameter (μm)
0.15	12	15	0.10	0.32 ± 0.07
0.15	12	15	0.18	0.39 ± 0.09
0.15	12	15	0.26	0.55 ± 0.10

**Table 6 molecules-24-03037-t006:** The optimal spinning conditions for three active molecules.

Active Molecule	PVP Addition Amount (g)	Voltage (kv)	Spinning Distance (cm)	Injection Speed (mm/min)
quercetin	0.20	16	20	0.26
glycyrrhizic acid:	0.15	12	15	0.18
α-mangostin	0.10	16	20	0.26

**Table 7 molecules-24-03037-t007:** The RSD for three active molecules with different methods.

Active Molecule	RSD% (UV.)	RSD% (FL.)	RSD% (CV.)
quercetin	0.184	0.304	0.291
glycyrrhizic acid:	0.686	0.487	0.384
α-mangostin	0.641	0.314	0.402

**Table 8 molecules-24-03037-t008:** The data of O_3_ etc. at the same stage of sampling for the detection of free radicals.

Active Molecular	Sampling Date	Sampling Time	AQI	T °C	Humidity %	PM_2.5_ μg/m^3^	PM_10_ μg/m^3^	O_3_ μg/m^3^	SO_2_ μg/m^3^	NO_2_ μg/m^3^	CO mg/m^3^	Reaction Capacity %	Free Radicals UV 10^7^/cm^3^	Free Radicals FL 10^8^/cm^3^	Free Radicals CV 10^7^/cm^3^
Quercetin	03/08	10:00–13:00	64	5.5	72.1	46	56	19	11	65	1.49	41.31	1.920 ± 0.410	4.588 ± 1.270	1.833 ± 0.459
		13:00–16:00	56	9.0	50.9	38	62	50	12	48	1.08	33.31	1.549 ± 0.035	3.182 ± 0.136	0.954 ± 0.420
	03/09	10:00–13:00	116	3.0	78	73	182	41	9.0	41	1.09	26.53	1.233 ± 0.280	3.361 ± 0.043	----
	03/11	10:00–13:00	60	3.2	50.2	26	70	68	21	62	0.90	32.88	1.529 ± 0.015	1.705 ± 1.613	----
	03/14	10:00–13:00	94	1.6	65.1	70	102	39	23	56	1.37	42.36	1.969 ± 0.455	4.242 ± 0.924	----
		13:00–16:00	80	10.7	37.1	59	66	86	19	30	1.07	30.57	1.421 ± 0.092	2.097 ± 1.221	----
	03/15	10:00–13:00	77	2.0	45.6	56	103	15	23	67	1.39	31.56	1.467 ± 0.046	3.790 ± 0.472	1.645 ± 0.271
		13:00–16:00	72	10.8	25.5	52	75	162	24	34	1.13	29.87	1.389 ± 0.124	3.078 ± 0.240	0.962 ± 0.412
		16:00–19:00	78	16.1	21.9	57	73	145	17	20	0.78	24.56	1.142 ± 0.371	3.818 ± 0.500	1.475 ± 0.101
Glycyrrhizic	03/02	13:00–16:00	108	8.8	55.3	81	103	90	16	45	1.56	11.53	0.593 ± 1.520	2.846 ± 0.621	2.210 ± 0.818
	03/06	13:00–16:00	85	8.8	35.5	56	120	79	28	40	1.39	3.80	0.196 ± 1.917	5.370 ± 1.904	2.135 ± 0.893
	03/11	10:00–13:00	60	3.2	50.2	26	70	68	21	62	0.90	52.32	2.689 ± 0.576	5.315 ± 1.849	4.043 ± 1.014
		13:00–16:00	53	9.9	33.3	23	55	100	15	34	0.58	42.33	2.175 ± 0.062	3.944 ± 0.478	4.537 ± 1.508
		16:00–19:00	44	13.5	20.3	23	42	140	16	17	0.53	75.67	3.889 ± 1.776	1.902 ± 1.565	2.217 ± 0.811
	03/16	10:00–13:00	105	4.6	51.2	79	118	23	19	61	1.37	61.01	3.136 ± 1.023	1.422 ± 2.045	----
α-Mangostin	03/02	10:00–13:00	107	2.3	76.3	80	117	31	17	66	1.81	21.43	1.101 ± 0.125	2.602 ± 0.162	----
		13:00–16:00	108	8.8	55.3	81	103	90	16	45	1.56	46.64	2.396 ± 1.421	1.850 ± 0.590	----
	03/06	10:00–13:00	77	3.2	49.4	56	90	22	24	69	1.47	19.59	1.007 ± 0.032	3.022 ± 0.582	----
		13:00–16:00	85	8.8	35.5	56	120	79	28	40	1.39	19.89	1.022 ± 0.047	2.950 ± 0.510	3.117 ± 0.246
	03/08	13:00–16:00	56	9.0	50.9	38	62	50	12	48	1.08	9.04	0.465 ± 0.510	1.494 ± 0.946	4.433 ± 1.070
	03/09	10:00–13:00	116	3.0	78	73	182	41	9.0	41	1.09	14.00	0.720 ± 0.255	1.877 ± 0.563	3.133 ± 0.230
	03/11	13:00–16:00	53	9.9	33.3	23	55	100	15	34	0.58	11.36	0.584 ± 0.391	1.273 ± 1.167	2.947 ± 0.416
		16:00–19:00	44	13.5	20.3	23	42	140	16	17	0.53	9.88	0.508 ± 0.467	4.451 ± 2.011	3.185 ± 0.178

**Table 9 molecules-24-03037-t009:** The chemical formula of natural active molecules.

Natural Active Molecules	Chemical Formula	References
quercetin	C_15_H_10_O_7_	[38,39]
glycyrrhizic acid	C_42_H_62_O_16_	[40,41]
α-mangostin	C_24_H_26_O_6_	[42]

**Table 10 molecules-24-03037-t010:** The separation conditions of three active molecules.

Subject	Solvent	Reflow Temperature	Solid-Liquid Ratio	Extraction Times	Extraction Time	References
quercetin	90%methanol	80 °C	1:10	3	2 h	[43]
glycyrrhizic acid	80%methanol	80 °C	1:8	3	2 h	[44]
α-mangostin	70%methanol	70 °C	1:10	3	2 h	[45]

**Table 11 molecules-24-03037-t011:** Single factor spinning parameter design.

PVP Addition Amount (g)	Methanol Solvent (mL)	Voltage (kv)	Spinning Distance (cm)	Injection Speed (mm/min)
0.10	5	10	10	0.10
0.15	5	12	15	0.18
0.20	5	16	20	0.26

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
