# Peer review of "Preparation of Electrospun Active Molecular Membrane and Atmospheric Free Radicals Capture"

_molecules, 2019, doi:10.3390/molecules24173037_

Round 1

Reviewer 1 Report

Dear authors,

Follows some comments to improve the manuscript quality:

- Introduction: is not clear if there are other metthods available , disadvantages, the advantage of use electrospinning. There are some capital letter in the middle of the sentences. Line 52: the fine stream...is the solvente? related to the electrospinning some technical terms are not being use correctly.   'spraying process' is spinning, no? 'finally falls' is deposited on the collector. miss some references.

- line 71: 'chemical reagent' PVP is a polymer.

- line 87: voltage can be high or low not large;

- line 102: Receive distance is not the correct technical name...is distance between tip of the needle and collector;

- line 137: '15cm' the optimum value is 20!??

- line 139: '0.18 mm/min' the optimum value is 15!??

- line 152-161: the discution must be improved, be careful, image B1 presents drops and C1 presents beads;

- Figure 6: I cannot understand how you did this, there are no explanation in the section 2;

- line 249: usually is posite or negative and ground...you described as positive and negative, are correct?

- line 253: voltage is only v and not kV?

- Table 2: the caption must contain the nomenclature, otherwise is difficult to the reader understand;

General: some english mistakes, capital letters, nomenclature description repetition, some images with partial cuts, no statistical analysis, no SD, some analysis performed are not described in the materials and methods section, confusition with verb tenses, conclusion must be improved.

Author Response

Response to Reviewer 1 Comments

Response to referees,

We thank the referees very much for your professional comments and great efforts on our work. We are deeply grateful to all the helpful suggestions. The manuscript has now been revised according to your suggestions in detail. The responses to your comments are listed below point by point. The corresponding revisions on the attachment manuscript has been marked with yellow font in the file of “the revised manuscript-marked”. We also checked the grammar and sentences, and necessary changes were made. We have highlighted all the changes with red font in the revised manuscript.

Comments from reviewer 1:

- Introduction: is not clear if there are other metthods available , disadvantages, the advantage of use electrospinning. There are some capital letter in the middle of the sentences. Line 52: the fine stream...is the solvente? related to the electrospinning some technical terms are not being use correctly. 'spraying process' is spinning, no? 'finally falls' is deposited on the collector. miss some references.

Response: Thank you very much for your professional guidance. We are very sorry for our carelessness and lack of knowledge. According to your guidance, we have modified them in the revised manuscript. We added some other methods in Line 47 and some disadvantages, the advantage of use electrospinning. In Line 52, we have modified some technical terms and added some references.

- line 71: 'chemical reagent' PVP is a polymer.

- line 87: voltage can be high or low not large;

-line 102: Receive distance is not the correct technical name...is distance between tip of the needle and collector;

Response: Thanks for your professor comment. We have studied these comments carefully and tried our best to revise and improve the manuscript. According to your guiding, we have revised the sentence in the revised manuscript.

- line 137: '15cm' the optimum value is 20!??

- line 139: '0.18 mm/min' the optimum value is 15!??

Response: Thanks for your professor comment. According to your guiding, we have summarized the optimized values in a Table 6. It will be helpful for the reader to follow easier the following discussion.

- line 152-161: the discution must be improved, be careful, image B1 presents drops and C1 presents beads;

Response: Thanks for your professor comment. According to your guiding, we redid a set of SEM of B1, C1.

- Figure 6: I cannot understand how you did this, there are no explanation in the section 2;

Response: Thanks for your professor comment. According to your guiding, we've deleted it, because this picture is not very relevant to this paper.

- line 249: usually is posite or negative and ground...you described as positive and negative, are correct?

- line 253: voltage is only v and not kV?

- Table 2: the caption must contain the nomenclature, otherwise is difficult to the reader understand;

Response: Thanks for your professor comment. According to your guiding, we have revised the sentence in the revised manuscript.

General: some english mistakes, capital letters, nomenclature description repetition, some images with partial cuts, no statistical analysis, no SD, some analysis performed are not described in the materials and methods section, confusition with verb tenses, conclusion must be improved.

Response: Thanks for your professor comment. We have studied comments carefully and have made correction which we hope meet with approval.

We appreciate Editors/Reviewers' warm work earnestly, and hope the correction will meet with approval. Once again, thank you very much for your comments and suggestions.

Reviewer 2 Report

The manuscript deals with the fabricationof electrospun mats containing active molecules able to capture free radicals from the atmosphere. The paper can be considered for publication after addressing the following issues:

It is not clear for the readers why the use authors used only clearance as method for the optimization of the electrospinning parameters and they never reported any morphological analysis (SEM) for the selection of the parameters. In fact, the morphology shown in Figure 5 for the “optimal” samples, actually it seems to not be the “optimal” because some beads are present in the mats, in particular for glycyrrhizic and mangostin samples (Figure 5B1 and 5C1). Please, explain it and if possible add SEM characterization for the measured clearance values.

Through the manuscript the information about the average fiber diameter seems to be missing. The readers could be interested also in knowing more about the filtering capability and the average fiber diameter of the mats. Please, add the measurements and discuss it.

Page 2 line 54: add “rheological” properties of the spinning solution

Page 2 line 71-73: Please reformulate the sentence, it seems not clear.

Page 2 line 77: It is not clear the “addition amount” of what in what, please specify better.

Page 3 line 87: substitute “larger” with “higher”.

Page 3 line 93-94; page 3 line 109-110; page 4 line 126-127: Please, clarify if the tested parameter values (of voltage, distance and flow rate) were selected based on previous literature (and in this case add the related references) or if were selected for specific reasons.

Page 4 line 136-142: please, summarized the optimized values in a table, this will be helpful for the reader to follow easier the following discussion.

Page 4 line 162: Paragraph 2.2.2. and the following Figure 6, seem to be not really related to the rest of the content, please clarify it or remove it.

Page 7 Figure 7: please add the error bars in Figure 7d.

Page 13 line 247-248: please, check the unit used for the needle. In fact, “needle No 21” is not acceptable and it is missing the unit, moreover for the outer diameter “0.8*80mm” is not the standard way to express it, please check and correct it.

Page 13 Table 3 and related text: The parameters related to the solution are not clear. In fact, the final concentration of the solution is not reported, it is also not clear how much PVP and how much active molecules were put in the solution. The solvent(s) were also not mentioned.

Page 13 line 259: could you please add references of the parameters (conditions) used for the test?
